# Development of an intervention to improve access to living-donor kidney transplantation (the ASK study)

**Pippa K. Bailey**[1,2]*, **Yoav Ben-Shlomo**[1], **Fergus J. Caskey**[1,2], **Mohammed Al-Talib**[1,2], **Hannah Lyons**[1,3], **Adarsh Babu**[4], **Liise K. Kayler**[5], **Lucy E. Selman**[1]

**1** Bristol Medical School, Population Health Sciences, University of Bristol, Bristol, United Kingdom, **2** Southmead Hospital, North Bristol NHS Trust, Bristol, United Kingdom, **3** Leeds School of Medicine, University of Leeds, Leeds, United Kingdom, **4** Gloucestershire Royal Hospital, Gloucestershire Hospitals NHS Foundation Trust, Gloucester, United Kingdom, **5** Erie County Medical Center, Buffalo, NY, United States of America

\* pippa.bailey@bristol.ac.uk

## Abstract

A living-donor kidney transplant (LDKT) is one of the best treatments for kidney failure. The UK's LDKT activity falls behind that of many other countries, and there is evidence of socio-economic inequity in access. We aimed to develop a UK-specific multicomponent intervention to support eligible individuals to access a LDKT. The intervention was designed to support those who are socioeconomically-deprived and currently disadvantaged, by targeting mediators of inequity identified in earlier work. We identified three existing interventions in the literature which target these mediators: a) the Norway model (healthcare practitioners contact patients' family with information about kidney donation), b) a home education model, and c) a Transplant candidate advocate model. We undertook intervention development using the Person-Based Approach (PBA). We performed in-depth qualitative interviews with people with advanced kidney disease (n = 13), their family members (n = 4), and renal and transplant healthcare practitioners (n = 15), analysed using thematic analysis. We investigated participant views on each proposed intervention component. We drafted intervention resources and revised these in light of comments from qualitative 'think-aloud' interviews. Four general themes were identified: i) Perceived cultural and societal norms; ii) Influence of family on decision-making; iii) Resource limitation, and iv) Evidence of effectiveness. For each intervention discussed, we identified three themes: for the Norway model: i) Overcoming communication barriers and assumptions; ii) Request from an official third party, and iii) Risk of coercion; for the home education model: i) Intragroup dynamics; ii) Avoidance of hospital, and iii) Burdens on participants; and for the transplant candidate advocates model: i) Vested interest of advocates; ii) Time commitment, and iii) Risk of misinformation. We used these results to develop a multicomponent intervention which comprises components from existing interventions that have been adapted to increase acceptability and engagement in a UK population. This will be evaluated in a future randomised controlled trial.

**Data Availability Statement:** This study generated qualitative data in the form of digital audio recordings and transcripts from interviews. Participants were asked to provide written consent

to share their anonymised data with other researchers. Data will be shared only if consent has been provided. 29 of 32 participants provided consent for data sharing. Anonymised interview transcripts have been uploaded to the University of Bristol's Research Data Repository: https://data.bris.ac.uk/data/. Audiofiles of the recorded interviews are not suitable for sharing as they carry a high risk of allowing the research participant to be identified, and the content of interviews includes sensitive information. Individuals who wish to access the dataset can contact the researchers directly or actively search the University of Bristol's data repository. Although the qualitative transcripts have been anonymised, as personal and sensitive issues have been discussed we cannot rule out the risk of identification, and therefore access to these transcripts is controlled. Individual researchers will need to request access to the controlled data through the University of Bristol via the Data Access Committee (DAC) for approval, before data can be shared after their host institution has signed a Data Access Agreement. The procedure for accessing data can be found here: https://www.bristol.ac.uk/staff/researchers/data/accessing-research-data/.

**Funding:** This research was funded in whole by the Wellcome Trust (PKB is funded by a Wellcome Trust Clinical Research Career Development Fellowship (214554/Z/18/Z) https://wellcome.org). LES is funded by a National Institute for Health Research (NIHR) Career Development Fellowship (https://nihr.ac.uk). MAT is funded by an NIHR Academic Clinical Fellowship (https://nihr.ac.uk). YBS is the cross-cutting methodological theme lead for the NIHR Applied Research Collaboration West (ARC West) (https://arc-w.nihr.ac.uk/). The animations evaluated were developed by LKK in USA Health Resources and Services Administration (HRSA) funded work (Grant No. R39OT31887, https://www.hrsa.gov). No funding body had any influence on the conception, design, analysis, or production of this manuscript, and the views it expresses are not necessarily those of any funding organisations. The views expressed in this publication are those of the authors and not necessarily those of the NHS, the Wellcome Trust, the HRSA, the National Institute for Health Research or the Department of Health.

**Competing interests:** The authors have declared that no competing interests exist.

# Introduction

A living-donor kidney transplant (LDKT) describes a transplant in which a kidney has been donated from a living person, typically a relative or friend. It is the best treatment in terms of life-expectancy for most people with kidney failure [1,2]. The risks of donating a kidney are small [3–5] and the quality of life of donors usually returns to pre-donation levels after donation [6,7].

The UK's LDKT activity falls behind that of many other countries, including the Netherlands and the USA [8]. Only 20% of those listed on the UK's kidney transplant waiting list receive a LDKT each year [9], and certain groups of individuals with kidney disease appear to be less likely to receive a LDKT. Socioeconomic deprivation describes the disadvantage of an individual or group relative to others in society, as indicated by people's education, employment, income, and assets [10,11]. In the UK the most socioeconomically-deprived people with kidney disease are 60% less likely to receive a LDKT than the least deprived [12]. Improving equity in living-donor kidney transplantation has been highlighted as a research priority by patients and clinicians [13,14]. In this study we aimed to develop a multicomponent intervention to support those currently disadvantaged in accessing a LDKT and to increase LDKT numbers in the UK. This study follows our previous mixed-methods research to identify barriers to living-donor kidney transplantation, and to understand reasons for the observed socioeconomic inequity in the UK (Fig 1).

Four variables have been identified as key mediators of socioeconomic inequity (Fig 2). Patient activation describes the knowledge, skills and confidence a person has in managing their own health and health care [15]. Social support comprises the emotional, physical, practical, informational, and relational assistance available to a person [16]; perceived social support describes what support an individual perceives is available and may not correlate with true available social support. Finally, health literacy describes an individual's ability to obtain, process, and understand basic health information needed to make appropriate health decisions [17]. Socioeconomic deprivation is associated with a lack of LDKT knowledge [18,19], lower levels of patient activation [18,19], perceived low levels of social support [18,19], and lower health literacy [20] (Fig 2).

## Required intervention components

The four factors identified above are potential targets for intervention. An intervention to overcome the identified barriers to living-donor transplantation therefore needs to include the components outlined in Table 1, to address barriers directly or provide a 'work-around' approach to them. For example, if low levels of patient activation mean that a patient find it difficult to approach potential donors, a 'work-around' solution would be for a healthcare practitioner to approach potential donors on a patient's behalf.

## Existing interventions

We investigated the existing evidence base to identify intervention components that might overcome the identified barriers. A recent scoping review summarised strategies to increase LDKTs, and concluded that there was an important gap in the literature for evidence-based interventions [21]. Two potentially delivered the required components:

• **Home-based patient and family education [22,23].** This was the only trialled intervention found to be effective in the scoping review [21]. It was developed with reference to Multisystemic Therapy theory [24] and trialled amongst disadvantaged populations in the USA [23] and the Netherlands [22]. Kidney patients and invited family members are visited at home by health workers who provide them with information on kidney disease, transplantation and

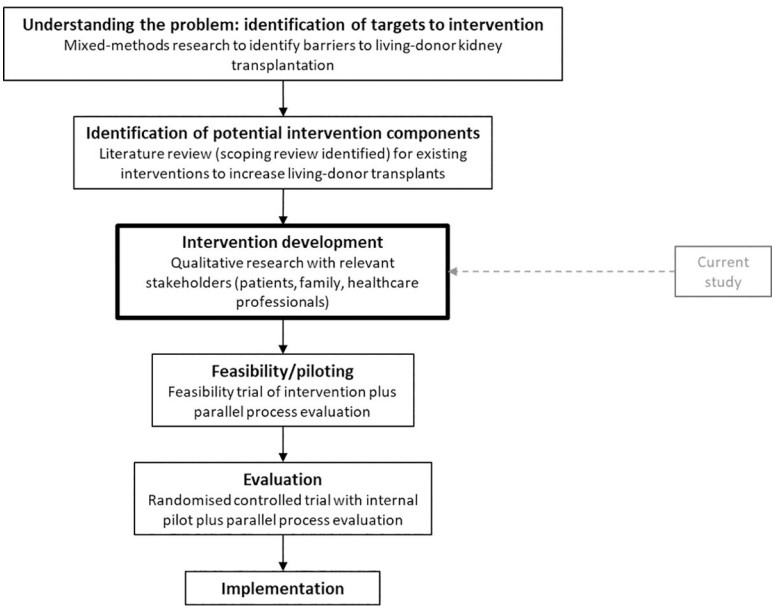

**Fig 1. Flow chart illustrating programme of research.**

donation, ensure patients know a LDKT is an option for them, engage their social network, and facilitate conversations about living kidney donation. Both trials reported >20% more LDKTs in the intervention versus control groups.

• **Transplant candidate advocates (TCAs) [25,26].** In this intervention a friend, relative or volunteer is trained as an advocate: someone willing to speak to other friends and family about donation on the patient's behalf. Although a small, single-centre observational

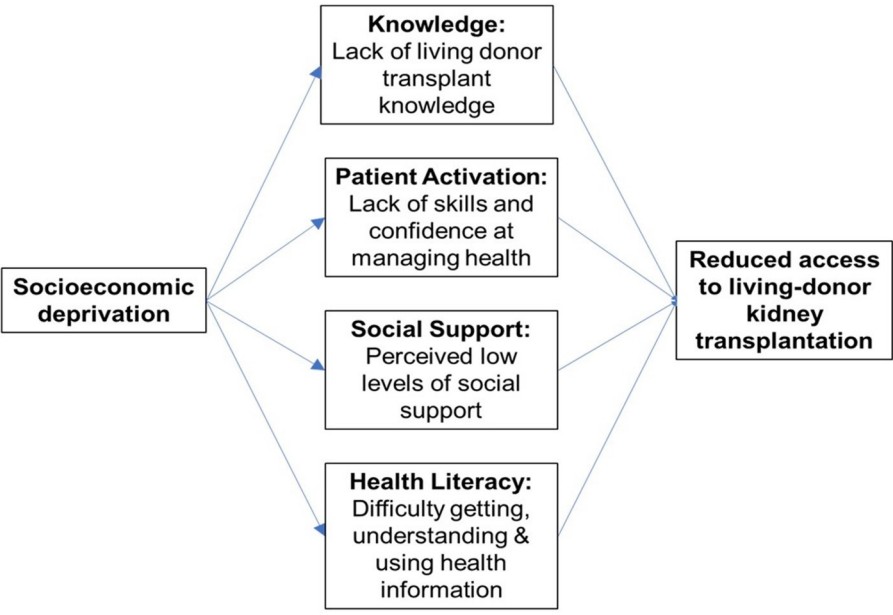

**Fig 2. Mediators of socioeconomic inequity in living-donor kidney transplantation.**

**Table 1. Intervention components required to address described barriers.**

| Required intervention components | Barrier addressed | | | |
|---|---|---|---|---|
| | Lack of knowledge | Lack of patient activation | Perceived low levels of social support | Limited health literacy |
| Informing kidney patients in a way tailored to those with limited health literacy of the personal option for them of a living-donor kidney transplant | X | X | | X |
| Identification of and healthcare practitioner engagement with the patient's social network | | X | X | X |
| Facilitation of conversations with potential donors | X | X | X | X |

evaluation of the intervention has been undertaken using a matched cohort study [25], there is currently no randomised controlled trial (RCT) evidence.

We also considered another intervention that targets the identified barriers, that is standard of care in Norway but has not been formally evaluated in an RCT:

• **Communication from healthcare provider to potential donors ('the Norway model').**   In Norway, people with advanced kidney disease provide their doctor with the details of friends and family members whom they are happy for the clinical team to contact with information on the person's need for a kidney transplant, and with information about living kidney donation [27].

We identified the above interventions as potential components of a multicomponent complex intervention.

The aim of this intervention development study was to develop and co-produce a UK-specific multicomponent intervention to support individuals eligible for kidney transplantation to access a LDKT. The intervention aims to target barriers particularly experienced by socioeconomically-deprived patients, but the intervention will be offered to all transplant candidates, in keeping with the concept of 'proportionate universalism' [28]. The intended intervention will be a complex intervention, in that it combines a number of interacting components [29]. We aimed to investigate the views of all relevant stakeholders on possible intervention components. We aimed to understand and accommodate the perspectives of the people who will both use and deliver the intervention, in order to improve acceptability, feasibility, engagement, and ultimately outcomes. The intended outcome of this work was a developed intervention to take forward into a clinical trial.

## Materials and methods

### Theoretical framework

We undertook intervention development using an approach that was both 'theory and evidence based' and 'target population centred' [30]. 'Theory and evidence based' approaches develop interventions by combining published research evidence and existing theories. As indicated above we identified three existing intervention components from the published literature. One intervention (Home-based patient and family education) had been developed with respect to multisystemic therapy theory and had RCT evidence of effectiveness. One intervention (TCAs) had weak evidence of effectiveness from an observational study, and one ('the Norway model') had not been formally evaluated in research. 'Target population centred' approaches develop interventions based on the views of the people who will use them, and we employed the 'Person-Based Approach' (PBA) to do this. The PBA [31] is a method for optimising intervention materials to ensure they are as acceptable and engaging as possible and feasible for use. Through qualitative interviews we aimed to understand how different people may view and engage with the proposed intervention components, which components seem

particularly relevant or attractive to them, and which may be rejected. The PBA involves the production of 'Guiding Principles' consisting of two elements: i) intervention design objectives, and ii) key features of the intervention that can achieve these objectives (Table 2).

## Study population

The study was undertaken at two UK hospitals (a transplanting centre and a transplant referring centre). Semi-structured qualitative interviews were undertaken with English-speaking UK-resident adults aged $\geq$ 18 years and <75 years, from the following groups:

- People with advanced kidney disease (including i) individuals with Chronic Kidney Disease stages 4 and 5, and ii) individuals receiving kidney replacement therapy (dialysis or a functioning kidney transplant)

- Family members of people with advanced kidney disease

- Healthcare practitioners who work with people with kidney disease

Individuals who lacked the Mental Capacity to consent to participation were not included. The eligible patient population was identified by the local site primary investigators. Individuals were invited to participate in a single face-to-face qualitative interview by i) post (invitation letter, patient information leaflet, a return response slip and pre-paid return envelope), ii) in person by a healthcare practitioner at a clinical appointment (e.g. in-centre haemodialysis session), and iii) through posters in outpatient clinics and haemodialysis units. Purposive sampling of patient participants was undertaken to ensure diversity in sex, age, ethnicity, and

**Table 2. Guiding Principles for intervention development: Intervention design objectives and key features of the intervention that can achieve these objectives.**

| Intervention design objectives | Key features of the intervention–detailing the characteristics of the intervention which address the objectives |
|---|---|
| i) To increase LDKT knowledge amongst people with kidney disease and their social network | • Informing people with kidney disease and their social network of the option of a living-donor kidney transplant (LDKT).<br>• Provision of information using multiple formats (face-to-face meetings, simple-language/Plain English written information, animations).<br>• Dedicated discussion about LDKTs with a specialist healthcare practitioner separate to usual kidney clinic consultation. |
| ii) To increase an individual's level of patient activation, or provide a 'work-around' solution | • Dedicated discussion about LDKTs with specialist healthcare practitioner separate to usual kidney clinic consultation.<br>• Healthcare practitioner assistance in the identification of individuals potentially eligible for donation from patient's social network.<br>• Direct engagement by healthcare practitioners with patient's social network and potential donors, with the patient's consent. |
| iii) To engage directly with an individual's social support network, including potential donors | • Direct engagement by healthcare practitioners with patient's social network and potential donors, using multiple formats (face-to-face meetings, simple language written information, animations). |
| iv) To tailor information to individuals with limited health literacy | • Any written information to be in simple language/Plain English tailored to individuals (patient and potential donors) with limited health literacy<br>• Provision of information using multiple formats (face-to-face meetings, simple language written information, animations). |

socioeconomic status using the following socioeconomic measures: patient's education level, employment status and housing/postcode). Demographic data on socioeconomic status was collected at interview. Family members were recruited via posters in hospital outpatient areas and through 'snowball sampling' through participants with kidney disease. Healthcare practitioners were invited to participate by the Chief Investigator (PKB) via email. Healthcare practitioners were purposively sampled to ensure diversity in sex, age, ethnicity and clinical role. All participants were aware of living-donor kidney transplantation as a theoretical treatment option for advanced kidney disease.

## Data collection

Interviews were undertaken by the Chief Investigator (PKB). In the interviews conducted at the beginning of the study, PKB discussed each proposed intervention component in turn (S1 File. Example topic guide). As the study progressed and the intervention became focussed on two components (see Results), intervention resources were drafted for these intervention components. The drafted intervention resources included:

i. **a letter from a healthcare provider to a potential donor** outlining the individual's need for a transplant, detailing the option of living kidney donation, and providing details on how interested individuals could find out more;

ii. **a simple language information leaflet on living kidney donation** entitled 'Donating one of your kidneys' and adapted from the Kidney Care UK leaflet 'Living Kidney Donation';

iii. **informational animations** on donation and the process of donation, developed by Dr Liise Kayler in Health Resources and Services Administration (HRSA) funded work in the USA [32].

The content of the proposed home education session was also discussed. During the qualitative interviews participants were asked to 'think-aloud' [33] as they reviewed the drafted intervention resources. The intervention resources were progressively modified in response to comments made in the interview to improve acceptability, comprehension, clarity, intelligibility, use and reach.

Interviews were either undertaken face-to-face or over the telephone. Face-to-face interviews were undertaken at a location of the participant's choosing (own home or hospital). Written consent was provided at the time of face-to-face interviews. For telephone interviews oral consent was recorded and written consent was confirmed via post. Participant demographic data were collected at interview (sex, 10-year age group, ethnicity, marital status, highest education level, employment status). A £20 voucher for participation was given to all participants.

## Analysis

Interviews were audio-recorded, transcribed verbatim, and analysed using inductive thematic analysis [34], as described by Braun and Clarke [35]. Anonymised transcripts were uploaded to NVivo software for analysis. All transcripts were coded by PKB and a subset were dual-coded independently by two other researchers (MA-T, an Academic Clinical Fellow in renal medicine, and HL, a Masters of Health Sciences Research student). Data collection and analysis were conducted concurrently, employing an iterative approach. The sample size was determined by reaching theoretical theme saturation [36].

Changes to interventions were made with reference to the Guiding Principles as outlined in Table 2. The criteria used for deciding whether to make a change to the intervention are

shown in Table 3. Changes were made if they were likely to impact on behaviour change or a precursor to behaviour change (e.g. acceptability, feasibility, persuasiveness, motivation, engagement) and were prioritised based on the MoSCoW (Must have, Should have, Could have, Would like) criteria [37,38] (S2 File. MoSCoW criteria).

If suggested changes compromised the Guiding Principles they were not implemented. Consensus was not a necessary goal of the intervention development process if differing preferences regarding content, delivery, and process could be accommodated (e.g. certain individuals may decline written material but accept links to animations).

We received NHS Research Ethics Committee (REC) (REC reference 19/WM/0320) and Health Research Authority (HRA) approval. The study was funded by a Wellcome Trust Clinical Research Career Development Fellowship (214554/Z/18/Z). The clinical and research activities being reported are consistent with the Principles of the Declaration of Istanbul as outlined in the 'Declaration of Istanbul on Organ Trafficking and Transplant Tourism'. The report was prepared with reference to the Guidance for reporting intervention development studies in health research (GUIDED) checklist [39] (S3 File. GUIDED checklist).

## Results

33 (36%) of 92 invited individuals agreed to participate (Table 4) but one individual was then unable to continue. Interviews ranged from 13–74 minutes in length with a mean duration of 42 minutes. One of the family members interviewed was related to one of the participants with advanced kidney disease. One family member was also a healthcare practitioner.

The themes identified are illustrated in Fig 3. Four general overarching themes were identified with respect to the proposed interventions, relevant to all intervention components. These were i) Perceived cultural and societal norms; ii) Influence of family on decision-making; iii) Resource limitation, and iv) Evidence of effectiveness.

For each intervention discussed, three themes were identified:

- Norway model: i) Overcoming communication barriers and assumptions; ii) Request from an official third party, and iii) Risk of coercion.

- Home education model: i) Intragroup dynamics; ii) Avoidance of hospital, and iii) Burdens on participants

- Transplant candidate advocates model: i) Vested interest of advocates; ii) Time commitment, and iii) Risk of misinformation

Participant demographics are presented alongside quotes: ethnicity and marital status are not included as these made participants identifiable.

**Table 3. Criteria for deciding whether to make a change to the intervention components.**

| Criteria | Means |
|---|---|
| Important for outcome | The change is likely to impact outcome or a precursor to outcome (e.g. acceptability, feasibility, persuasiveness, motivation, engagement). |
| Consistent with Guiding Principles | The change is in line with the Guiding Principles of the intervention. |
| Consistent with Common Guiding Principles | The change is in line with common Guiding Principles: to support autonomy, promote competence and provide a positive emotional experience and sense of relatedness |
| Uncontroversial and easy | An uncontroversial and easy to implement solution that doesn't involve major design changes e.g. simplifying a sentence or replacing a word. These changes were implemented immediately. |
| Repeated by several participants | The point was made by more than one participant. |

**Table 4. Participant characteristics.**

| | n = 32 |
|---|---|
| **Characteristics** | **Number (%)** |
| **Sex** | |
| Female | 17 (53) |
| Male | 15 (47) |
| **Age group (years)** | |
| 20–39 | 4 (13) |
| 40–59 | 23 (72) |
| 60–79 | 5 (16) |
| **Ethnicity** | |
| White | 27 (84) |
| Other ethnic groups[1] | 5 (16) |
| **Marital status** | |
| Single | 9 (28) |
| Married/Long-term partner | 20 (63) |
| Other (Divorced or widowed/bereaved) | 3 (9) |
| **Participant group** | |
| People with advanced kidney disease[1] | 13 (41) |
| Family members[2] | 4 (13) |
| Healthcare practitioners | 15 (47) |
| • Transplant nurses and coordinators | 5 (33)[3] |
| • Home dialysis nurses | 3 (20)[3] |
| • Nurse other e.g. ward, haemodialysis | 4 (27)[3] |
| • Transplant physicians/surgeons | 3 (20)[3] |
| **Patients and family—highest level of education** | **n = 17** |
| Secondary school | 1 (6)[3] |
| Vocational/Technical training | 6 (41)[3] |
| University undergraduate degree | 2 (12)[3] |
| University postgraduate degree | 4 (24)[3] |
| Not disclosed | 3 (18)[3] |
| **Patients and family—employment status** | **n = 17** |
| Unemployed | 8 (47)[3] |
| Full or part-time employment | 4 (29)[3] |
| Retired and other (e.g. student, homemaker)[1] | 4 (24)[3] |

[1] Unable to provide information on subgroups due to small numbers in groups risking identification.

[2] One family member was also a healthcare practitioner. They are included here as a family member.

[3] % of 17 subgroup sample not % of 32 total sample.

## General themes

**Perceived cultural norms.** When discussing the proposed interventions, participants made reference to 'cultural' norms. They described how some aspects of the interventions were not yet established or accepted cultural norms. Some intervention components were seen as 'of another culture', and needed time for adoption in the UK:

> *(With respect to the home education model) 'I just think in Britain we would really struggle to set it up, I just think that it's not the done thing to sit in your lounge and discuss kidney disease . . . That's just stiff upper lip British.' (Transplant nurse or coordinator/Female/50-59 years).*

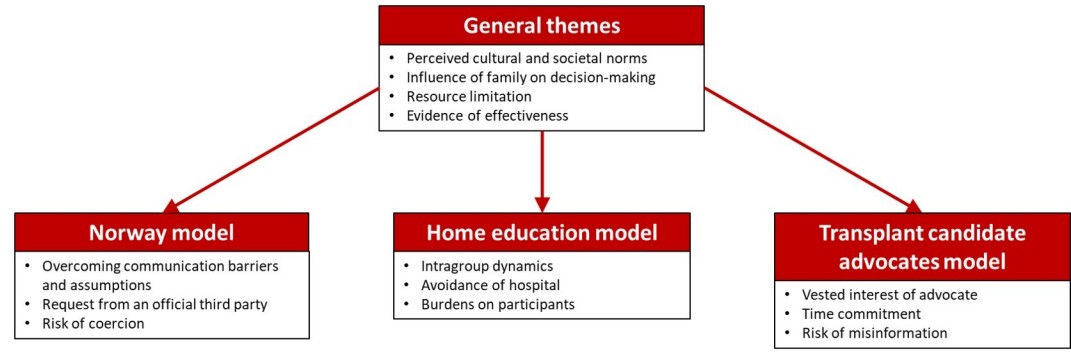

**Fig 3. Thematic diagram.**

*(With respect to Transplant Candidate Advocates) 'You see so many things from over in America where people stand out in the road with a sign, hit Facebook, and I'm not sure how I feel about that . . . to me that's like begging.' (Patient/Female/50-59 years/Full or part-time employment).*

Participants discussed how normalisation of the interventions was important. This required adoption of the intervention as 'standard, official practice' which in turn would make any intervention feel less targeted or coercive:

*'Participant: I'm interested that that's improved uptake, I wonder if that's a particular population thing? Maybe with other people that would work really well.*

Interviewer: But not for you?.

*Participant: I don't think so. But then if it was a cultural norm maybe it would be. Maybe it would be something that people just did.' (Family member/Female/30-39 years/Full or part-time employment).*

One healthcare worker explained how normalisation would take time:

*'I think if we do introduce something like this, yeah loads of people will have issue with it, but in ten years' time it will be normal, in 20 years' time, in 30 years' time it will have been forgotten before we did X, Y and Z.' (Transplant nurse or coordinator/Female/50-59 years).*

**Influence of family on decision-making.** Several participants emphasised the role and influence family members have on decision-making regarding living kidney donation. All three intervention components discussed involve direct engagement with a transplant candidate's family.

*'. . .it's not only me going to decide [about kidney donation], obviously my family, my wife, my daughter and 'Are you really sure want to do that [donate a kidney]?" (Home dialysis nurse/Male/50-59 years).*

*'I'm not prepared for my children to be a donor, I will leave [Name redacted] if he asks, like, I cannot do that.' (Family member/Female/30-39 years/University postgraduate degree).*

*[Brother withdrew offer to donate] 'So I think she [his brother's wife] had taken over the decision making from my brother and that did kind of hurt a little bit.' (Patient/Male/30-39 years/Unemployed).*

**Resource limitation.**   Many participants, particularly healthcare workers, expressed concern about the resources required for intervention delivery. They queried whether the NHS had resources to deliver the interventions and to respond to an increase in living donor enquiries resulting from the interventions if effective:

'*It would be very resource intensive and I can't imagine that we would ever be in a position to be able to do that.*' *(Nurse other/Female/40-49 years).*

'*Well the barriers would be I suppose finance, resources, you know . . . if it wasn't for the fact that it was labour intensive I'd say that would be fantastic.*' *(Family member/Female/60-69 years/Retired).*

'*So it's a land of milk and honey isn't it? . . . I think that's fantastic, but how you can possibly roll it out with resources?*' *(Transplant physician or surgeon/Male/40-49 years).*

One healthcare worker suggested existing healthcare staff would not have the capacity to deliver the programme, suggesting dedicated resources were required to sustain a service:

'*I think given the current pressure that we are under, if you asked people on our teams to give a very bespoke tailored, out of hours kind of approach to education in this area, I think you would probably find you wouldn't be able to sustain it.*' *(Home dialysis nurse/Male/30-39 years).*

**Evidence of effectiveness.**   Many participants, particularly healthcare practitioners, asked whether the interventions being discussed had been shown to be effective, something that was seen as important before a similar approach was adopted in the UK.

'*As an idea it sounds brilliant. May I ask, do you know how successful that is?*' *(Patient/Male/60-69 years/Unemployed).*

'*Does it yield benefit?*' *(Nurse other/Female/50-59 years).*

One participant reported that evidence of effectiveness may be important to help convince healthcare centres and practitioners to adopt the intervention.

## Norway model

The majority of interviewees across all participant groups responded positively to the suggestion of the 'Norway model' involving direct engagement of a transplant candidate's social network through the distribution of letters and information from healthcare providers.

All interviewees except one reported that this intervention component was a good idea and would be welcomed by them:

'. . . *it's just highlighting the issue I guess and it's still giving them a choice and that's basically what's important to me I think, is not to be too overpowering and to give them the choice. . .*' *(Patient/Male/30-39 years/Unemployed).*

'*I think it's a great idea to send that letter out with the details, because unless people get to hear about it . . . it would save sending a letter like that to maybe I don't know twenty people that would be on my list, if it was that many, would be a lot easier than me ringing twenty people.*' *(Patient/Male/60-69 years/Unemployed).*

One healthcare practitioner described how it also normalised the process for potential donors, which took away a sense that one individual was being targeted for donation:

'*As it goes to everyone, it takes away the individual target which perhaps makes people feel uncomfortable. . .' (Nurse other/Female/40-49 years).*

One patient participant reported that this intervention component wouldn't help them, but recognised that the approach might benefit others:

'*. . . all these impersonal approaches I wouldn't be in favour of. But that's only because maybe I'm comfortable having that conversation with somebody.' (Patient/Male/40-49 years/Full-time employment).*

**Overcoming communication barriers and assumptions.** Healthcare practitioners reported that it was important to ensure potential donors had been informed of the opportunity to donate. They reported that the letter and information sheet may help to do this and address any incorrect assumptions about donation:

'*. . .it might just raise awareness if nothing else and it might just give them that door to open to say, 'I didn't know that was something I could consider doing.' I think personally there might be a myth that if you were over a certain age you couldn't put yourself forward so people possibly rule themselves out because they think, 'I'm too old, I'm not fit enough." (Nurse other/ Female/50-59 years).*

Participants from all groups reported that knowing the letter was being sent would be a trigger/springboard to them having a personal conversation with potential donors, facilitating these discussions:

'*If I'm honest, I'd want to ask them first. I think it would be easier to say, 'I've been to the clinic. You know I've got this wrong with me. This is what they've given me, some leaflets that I can give out to family members." (Nurse other/Female/50-59 years).*

**Request from an official third party.** The greatest perceived benefit was the involvement of a third party in approaching potential donors: patients reported that this removed feelings of selfishness and required less effort and time than a patient contacting all potential donors directly. The burden of responsibility and stress was removed by someone else asking on their behalf:

'*It's coming from a third party so in some ways . . . I would feel less selfish if that letter was being sent out. . .' (Patient/Male/70-79 years/Retired).*

The 'third party' being an official recognised professional (for example a medical practitioner) or organisation was seen as being of particular importance:

'*Just say someone like me, from my background, where I've lied and cheated a lot you know what I mean . . . I don't get a lot of credence for my views so from a doctor is definitely more . . . there's a lot more weight behind it.' (Patient/Male/40-49 years/Unemployed).*

In addition, the use of an official, standardised letter was also seen as 'normalising' the need for a kidney transplant for potential recipients:

'*I think it's a great idea. I think it puts it in the context that it isn't just me, it normalises i.t'* (*Patient/Male/70-79 years/Retired*).

The request coming from a third party was considered as allowing patients to distance themselves slightly from the request, potentially protecting a relationship and reducing any perceived direct personal pressure from the transplant candidate on a family member to donate:

'*It's just helping to bring up that conversation, so if we get a letter from a consultant, then you might phone your brother [hypothetical transplant candidate] and say, 'I have this letter. . .' then he can of course say, 'You don't need to, this is just the hospital doing this kind of letter'.'* (*Transplant nurse or coordinator/Female/40-49 years*).

'*I am confident that I could . . . put it into context that you know, this isn't pressure from anybody, this is just awareness raising but not just from me but from the system, you know?'* (*Patient/Male/70-79 years/Retired*).

**Risk of coercion.** Some individuals expressed concern that the letter could be coercive and put pressure on individuals to donate. Most concerns were expressed by healthcare practitioners:

'*I think having a letter could make you feel more uncomfortable because you would have to give your reasons why you wouldn't give a kidney, as opposed to picking up the leaflet and saying actually I would like to find out more, one is positive I would like to find out a bit more or actually no I am not in a place to do that, whereas with a letter you have almost got to say why, why you don't want to give a kidney.'* (*Transplant nurse or coordinator/Female/50-59 years*).

'*I think that is coercive. I think that personally is a step too far.'* (*Transplant nurse or coordinator/Female/40-49 years*).

Only two patient participants felt the intervention was potentially at risk of causing harm:

'*It's a bit too invasive, it's a bit too like sat outside with your begging bowl.'* (*Patient/Male/40-49 years/Unemployed*).

'*. . .it might be coercive, it depends, like I said, it depends on the individual and how they interpret the letter*: '*Am I being asked to do this or am I being nominated to do this*?" (*Patient/Male/40-49 years/Full-time employment*).

However, overall this intervention component was perceived by most participants as having a low risk of harm as the letter could be ignored:

'*It might just open up conversations, it might be a bit of paper that goes in the recycling bin. . .'* (*Transplant nurse or coordinator/Female/50-59 years*).

**Home education model.** The majority of participants were positive about the home education model.

'*I think it's a fantastic idea. . . . It gives that person the opportunity to possibly get a donor and for the information to be told amongst the family at the same time, so they all know the situation anyway.*' (Patient/Male/40-49 years/Unemployed).

'*Magnificent idea.*' (Patient/Male/70-79 years/Retired).

'*As an idea it sounds brilliant.*' (Patient/Male/60-69 years/Unemployed).

**Intragroup dynamics.** Participants suggested that group education was better than individual education due to intragroup dynamics enhancing the interaction. Participants reported that family presence might bolster confidence in asking questions:

'*Yeah and I think if you're actually on home surroundings . . . that you know and you feel comfortable with I think you're more open and, not direct, but you'll be open in asking questions that are on your mind. Rather than keeping them back and thinking about them and think, 'Well, maybe I should have asked that, I should have asked that'. Because there'd be no need to because you'd have support of their family, the family are there with you.*' (Patient/ Male/50-59 years/Full-time employment).

Two participants also suggested that other group members might ask questions you'd not considered, which might trigger other members to ask other related questions:

'*. . .the question you don't know you're gonna have might come out during that time and things like that.*' (Patient/Male/40-49 years/Full-time employment).

A few participants expressed concerns about negative intragroup dynamics:

'*. . . you may have a family where then you've got the partners are split up. Then you've got other partners. So, they all live in different houses and again there may be internal–you know–friction.*' (Nurse other/Female/50-59 years).

'*I've got some of my family members who are very quiet, and they'd just listen to all of it and wouldn't actually say anything. Then I've got others that will just take over the whole thing,*' (Nurse other/Female/50-59 years).

'*. . . there's that sort of peer pressure thing going on and perhaps that competitive, 'Oh they've offered, I'll offer' you know, particularly thinking about sort of young men and the, I don't know, testosterone-fuelled.*' (Family member/Female/30-39 years/Full or part-time employment).

One previous kidney donor highlighted that family group education lacked the intergroup dynamics that exist with group education of multiple families, suggesting that being educated in a small family group in the home meant the benefits of mixing with others with kidney disease and their families were lost:

'*Because there is something about if you've gone to the house and you meet the friends and family, okay it's still keeping whatever thoughts or beliefs are in that group. Whereas if you come out of your group and go to hospital you see a much broader cross section of people suffering and they've all got different questions and it broadens your mind to think outside your normal way of thinking.*' (Family member/Female/60-69 years/Retired).

**Avoidance of hospital.** Participants reported that both patients and their invited guests would feel more relaxed in the home, and reported that this created a more relaxed, social rather than formal atmosphere. Participants reported that this might encourage participation and engagement:

'*I think if somebody is at home, they feel more in control. It is their home and they feel more comfortable and relaxed.*' (Transplant nurse or coordinator/Female/50-59 years).

'*I think that's a big plus for everybody involved. It's kind of a social occasion rather than a too formal occasion. . .*' (Patient/Male/70-79 years/Retired).

'*. . . if it was me in that situation the fact that I've actually made the effort to go to somebody's house as a friend, means I'm more likely to engage with the person concerned than if I went to a hospital or a function room or something else.*' (Patient/Male/60-69 years/Unemployed).

One patient participant described how although they had become accustomed to the hospital environment, this was not true of their social network:

'*I think it would be more relaxing and more beneficial if it was in your own environment than in the hospital. Being in clinical surroundings is not always comfortable for people. It doesn't bother me, I've been doing it for too long.*' (Patient/Female/50-59 years/Unemployed).

As illustrated above, one of the perceived benefits of the home was the avoidance of a clinical environment in which people who have not become accustomed to it may feel uncomfortable. Many participants described how not needing to travel to a hospital meant avoiding the difficulties and costs with which this is associated, as well as the already described discomfort in clinical environments:

'*It's a long way to go from here to [Hospital name redacted] if you're not feeling well and [Participant's son] finds it very, very tiring.*' (Family member/Female/70-79 years/Full or part-time employment).

'*I mean the trouble with [hospital] is it's such a difficult hospital to get to and the parking is— before you've even sniffed you've got to pay £3.50.*' (Family member/Female/60-69 years/ Retired).

One healthcare professional highlighted that home visits may be particularly welcome in the context of the Covid-19 pandemic if people are worried about the risk of acquiring infections in the hospital environment:

'*I think that the hospitals are going to be stigmatised and I think home therapy and home visits are going to be really, really helpful.*' (Transplant physician or surgeon/Male/40-49 years).

**Burdens on participants.** Seven participants expressed negative opinions about the home-based education model. They reported that a home visit may be associated with additional burdens on the participants, both the hosts and guests:

'*I think it will be a pressure for them, if they are from a poor background with a hospital person coming. If you have a guest coming into your house you have the preparation and things, and then you have that unnecessary stress.*' (Transplant nurse or coordinator/Female/40-49 years).

The home visits were perceived as particularly burdensome for people who were in full-time employment:

'*How would they manage that*? *How would anyone with a full-time job come for a 7 o'clock at our house*? *The reality of contemporary life, we work full-time, everyone we know works full-time, in demanding jobs, how–and you're not even coming to socialise, you're coming to be lectured on something you know already to an extent, with the–we're hosting like a hustling session basically.*' (Family member/Female/30-39 years/Full or part-time employment).

## Transplant candidate advocates

The transplant candidate advocate intervention component was the least popular option, with most participants expressing negative views towards it or concerns about its use.

**Vested interest of advocates.** Several participants expressed concern that an advocate would have a vested interest in finding a potential donor, due to their potential to improve their own quality of life if an individual with whom they have a close relationship receives a transplant, as well as the potential to themselves avoid donation if they can find an alternative donor. Concern was expressed that such investments compromised the independence and impartiality of an advocate, and created potential for the advocate to be coercive or manipulative:

'*. . . if you're using somebody's husband as the advocate I think they've probably got a vested interest in getting that kidney and I think if you were going down that route I think I would have reservations.*' (Home dialysis nurse/Male/50-59 years).

One interviewee who was the spouse of a transplant candidate suggested requests to consider donation from one family member to another could be perceived as 'manipulative', and were better coming from a healthcare professional:

'*[Transplant candidate's name redacted]'s Dad could be asked by the nurse, 'you are best position given that you live in London and all your sisters are, you know, would you mind just scouting it out*?' *And that would have more gravitas than coming from me or from [Transplant candidate's name redacted] and seems less manipulative.*' (Family member/Female/30-39 years/Full or part-time employment).

As advocates could be invested in finding a potential donor in order to avoid donation themselves, two healthcare workers suggested that an advocate should have first been considered as a donor:

'*I'd find it weird if a family member came and asked me, before ruling himself or herself out.*' (Transplant physician or surgeon/Male/40-49 years).

**Time commitment.** Participants reflected on the time commitment required from the advocate to undergo training and to deliver this intervention. Potential advocates were suggested as having limited capacity to perform their required roles.

'*That's quite a big commitment for someone isn't it?*' (Transplant nurse or coordinator/ Female/40-49 years).

Participants considered their own possible advocates and discussed their lack of time due to work and caring responsibilities:

'*Yes, she's [potential advocate] got three or four kids and she's got to look after them, yes, and she works flat out.*' (Patient/Male/40-49 years/Unemployed).

**Risk of misinformation.**   Participants expressed concern about the quality of the information sharing by advocates who might lack the knowledge required to engage in conversations with possible donors. Participants raised concerns about the risk of misinformation, and of offence being cause by abrupt communication styles.

'*You don't actually know what they're going to say and you don't know if it's going to be appropriate or accurate.*' (Nurse other/Female/50-59 years).

One participant suggested the consequences of miscommunication or misunderstandings could be upset relationships between the transplant candidate, the advocate and the family and friends.

'*It is like Chinese whispers, you are going to say one thing . . . and then a different thing comes out altogether. It could actually end up being quite dangerous and maybe cause quite a lot of bad feeling if you have to say, 'that wasn't quite what I said' then, 'no but you gave me all those education sessions'. I think there is a risk. . . . if they get it wrong, it is awkward all round. You don't want bad feeling between someone that wants a transplant and the people that are trying to help them and that is what will happen. Then all that trust is lost.*' (Transplant nurse or coordinator/Female/50-59 years).

Although the majority of participants didn't think the advocate model was the best approach, some did express positive views. One patient reported that it would overcome personal communication barriers: he described himself not being someone who would ask anyone to consider donation, but his family members wouldn't hesitate to:

'*I'm pretty quiet, I wouldn't ask anyone do you know what I mean? So they'd [sister and mum as advocates] be banging down doors, they would, they'd be do it on my behalf, so it'd be helpful definitely.*' (Patient/Male/40-49 years/White/Unemployed).

One healthcare worker thought the role might be important for communities for which alternative roles such as home-based educators, may be undertaken by individuals who do not have the understanding or trust of a specific community. In this situation advocates who could communicate with such a community was viewed positively:

'*It's a really good idea . . . it might be that there could be someone that could help and approach their community who knows the culture, someone that they trust and is seen as significant within the community, but it's not attached to the hospital.*' (Transplant nurse or coordinator/Female/50-59 years).

The findings of the 'think-aloud' interviews with respect to the intervention resources and the MoSCoW decisions are presented in S1 Table. These findings are summarised in Table 5.

## Discussion

In this study, in conjunction with relevant stakeholders, we have developed a multicomponent complex intervention designed to improve access to living-donor kidney transplantation. The intervention comprises components from existing interventions that have been adapted to

**Table 5. Summary of changes to intervention components and resources (Summary of S1 Table).**

| Participant suggestions | Possible change(s) | Agreed change? |
|---|---|---|
| **Intervention resource: Invitation letter from healthcare practitioners to potential donors** | | |
| 1. Need for warning | 1. Encourage participants to tell family/friends to expect letter | Yes |
| 2. Written in English–excludes individuals who do not read English | 2. Translation of documents | No–cultural adaptation planned as later work |
| 3. Any written information risks excluding individuals with poor literacy | 3. Ensure language simple and letter short, and combine with other components as planned (eg. Face-to-face discussion, animations, home visit) | Yes |
| 4. Expected response needs to be clear | 4. Include sentences in letter making next steps clear | Yes |
| 5. Avoid targeting an individual | 5. Remove personalised aspects of letter i.e. Dear XXX | No–a Guiding Principle is to engage social network |
| **Intervention resource: Information leaflet on living kidney donation** | | |
| 1. Need for simple language | 1. Use simple language in leaflet e.g. replace urine with wee, replace cardiac with heart | Yes |
| 2. Section '*What tests will I need to give a kidney*?' too long | 2. Reduce the section entitled 'What tests will I need to give a kidney?'–currently across 2 pages–reduced to 1 page | Yes |
| 3. Lack of personal stories | 3. Add personal accounts of donation/transplant | Yes |
| 4. Statement that payment for donation is illegal unnecessary | 4. Remove section on payment for donation being illegal | No–important to highlight legal boundaries |
| **Intervention resource: Informational animations** | | |
| 1. Difficult to use as a reference | 1. Use in combination with written literature | Yes |
| 2. Need to be tailored for UK | 2.Change USA references to UK references and replace US voiceover with English voiceover | Yes |
| **Intervention resource: Home visit content** | | |
| 1. Content needs to be broad | 1. Education session to cover kidney disease, dialysis, transplantation and living donation | Yes |
| 2. Tailored to individual | 2. Tailor content with respect to primary disease, kidney replacement therapy options. | Yes |
| 3. Use professionals not patient educators | 3. Use of professional, trained home educators | Yes |
| 4. Use two educators–for safety/engagement | 4. Home visits to be undertaken by two home educators | Yes |

increase acceptability and engagement in a UK population. Intervention resources have been produced after an iterative development process. We have gained insights into which of the intervention components should be discarded, and which aspects of the intervention should be fixed or flexible in the full trial [40]. The intervention has thus been developed and optimised prior to evaluation in the ASK trial: improving AccesS to Kidney transplantation (ISRCTN registry reference ISRCTN10989132 https://doi.org/10.1186/ISRCTN10989132). The final intervention for trial is detailed in the Template for Intervention Description and Replication (TIDieR) checklist [41] in S2 Table.

The developed intervention may undergo further refinement following evaluation in the feasibility trial [42]. Through a mixed-methods process evaluation we will aim to understand people's experiences of and views on the intervention and resources at delivery in the real world, refining and adapting the intervention as necessary. We will also aim to understand how to optimise implementation and understand the influence of context on intervention delivery and effectiveness [30].

The initial proposed intervention compromised components that had already been delivered and/or evaluated in other populations in the Netherlands, the USA and Norway [22,23,25,27]. We found that interventions delivered in one population may require modification before being adopted in other populations. The Transplant Candidate Advocate intervention component, previously evaluated in observational work in the USA, was not popular with participants and this component will not be part of our intervention. This finding may be because individuals in this UK population have different views to those of individuals in a USA healthcare population. However, to our knowledge, no qualitative evaluation of the Transplant Candidate Advocate intervention has been undertaken in the USA, and therefore the concerns participants raised in this study may not be population specific. This might be more likely given that the identified themes were not related to UK culture or the healthcare model (Vested interest of advocates; Time commitment; Risk of misinformation). However, some participants perceived the intervention components as belonging to another culture, and therefore not an approach that translated to the UK. Changing cultural norms was seen as possible and participants suggested time would be the biggest factor in models of care becoming established practice.

The Theory of Planned Behaviour [43,44] describes how an individual's behaviour (e.g. to engage in the proposed intervention, to share information on living donation with family members, to invite guests to a home education session) is influenced by normative beliefs and perception of social norms. In this study, one participant indicated that knowledge that other people were engaging in these activities (*'something that people just did'*) may have change her own unwillingness to engage. The beliefs and attitudes of family members and friends contribute to perceived social norms. Our study findings emphasised the importance of the family on decision-making regarding donation and acceptance of a LDKT, something previously reported in numerous qualitative studies and a systematic review [45]. This finding highlights the importance of engaging with family members and friends even if they are unsuitable or unwilling to personally donate as they may still be influential on the decisions and actions of other potential donors and the transplant candidate, with capacity to affect the perceived social norms.

'Resource limitation' was an identified general theme, with participants expressing concerns about the resources required to deliver the intervention. The intervention was recognised as requiring dedicated resources, including professionals specifically employed to deliver the role. The process evaluation planned to run parallel to the trial will also allow us to evaluate both the capacity of the NHS to deliver the intervention and the impact of delivery on existing NHS services. This is essential to ensure an effective intervention is not rejected because of limited NHS resources, and to ensure an intervention effective at increasing LDKTs doesn't have a negative impact on another service. Related to appropriate allocation of limited resources was the question of whether there was evidence that the proposed intervention components were effective. No participants expressed strong beliefs that one intervention component would be effective, which suggests there is equipoise as to whether the interventions would work or not, important for recruitment to and delivery of an RCT [46]. In addition to effectiveness, given the significant required resources to deliver the intervention, cost-effectiveness information is crucial to guaranteeing healthcare funding for the intervention. A cost-effectiveness evaluation will be essential in the final full-scale trial, and information on cost-drivers will be evaluated in the feasibility trial.

## Strengths and limitations

This study provides an in-depth investigation of the views of people with kidney disease, family members, and healthcare workers towards potential interventions to improve access to living-

donor kidney transplantation. Talking participants through the intervention components in qualitative interviews has allowed different perspectives to emerge compared with responses to closed or more abstract questions [47]. Participants shared their views on both the proposed intervention components and drafted intervention resources, and the work achieved the aim of developing a multicomponent intervention and resources for delivery. Theme saturation was reached, and patient and healthcare worker participants were purposively selected for maximum diversity to capture the views of a variety of stakeholders. There are some limitations: i) Only 4 family members participated. Recruitment was limited by two main factors. Firstly, family members could be recruited by patient participants inviting their family members which may have been difficult and akin to approaching them to consider donation. Secondly, the Covid-19 pandemic meant that family members were not allowed to attend hospital appointments with patients and therefore exposure to waiting room posters about the study was limited. ii) Although participants were sampled from two NHS trusts they were from one region of the UK. Although a diverse sample was achieved, findings may not transfer to other regions or other centres in the UK. iii) Interviews were carried out by a clinician (PKB), known to most of the healthcare workers. Although participants spoke freely we are unable to determine if this altered responses. PKB was not known to any of the patient or family participants. iv) Interviews were not undertaken with individuals who did not speak English, and study resources have only been developed in English. Whilst resources could be translated we do not feel that this will be sufficient adaptation for individuals who do not speak English. This intervention has been developed to particularly address variables that mediate socioeconomic inequity and may not address barriers that explain ethnic inequity in access to living-donor transplantation [48], which differ from those being targeted here [49]. If the intervention proves acceptable, feasible and effective it will require formal adaptation for non-English speaking groups. This will require adaptation beyond simple translation of language, and will require cultural adaptation, as well as specifically addressing different barriers to living-donor transplantation. Translating intervention resources without additional cultural adaptation risks inappropriately discarding an effective intervention. Such cultural adaptation is required when evaluating an intervention in any new population: as illustrated in this study, interventions used in other countries do not automatically translate to a UK population.

## Conclusions

Improving equity in living-donor kidney transplantation has been highlighted as a UK and international research priority by patients and clinicians [13,14]. Through this study we have developed a multicomponent complex intervention, incorporating components developed for other populations. We have adapted and optimised the intervention and resources for use in a UK renal population, ready for evaluation in a feasibility and later full-scale randomised controlled trial. We have demonstrated that interventions delivered in one population may not be suitable for use in other populations without adaptation.

## Supporting information

**S1 Table. Changes to intervention components and resources.**
(DOCX)

**S2 Table. Final intervention as per the Template for Intervention Description and Replication (TIDieR) checklist.**
(DOCX)

**S1 File. Example topic guide.**
(DOCX)

**S2 File. MoSCoW criteria.**
(DOCX)

**S3 File. GUIDED checklist.**
(PDF)

## Acknowledgments

We would like to acknowledge two patient representatives from Bristol Health Partners' Kidney Disease Health Integration Team, Primrose Granville-McIntosh and Soumeya Bouacida, who contributed to the content and design of the study documents. The views expressed in this publication are those of the authors and not necessarily those of the NHS, the Wellcome Trust, the HRSA, the National Institute for Health Research or the Department of Health. For the purpose of Open Access, the author has applied a CC BY public copyright licence to any Author Accepted Manuscript version arising from this submission.

## Author Contributions

**Conceptualization:** Pippa K. Bailey, Fergus J. Caskey.

**Data curation:** Pippa K. Bailey, Adarsh Babu.

**Formal analysis:** Pippa K. Bailey, Mohammed Al-Talib, Hannah Lyons.

**Funding acquisition:** Pippa K. Bailey.

**Investigation:** Pippa K. Bailey.

**Methodology:** Pippa K. Bailey.

**Project administration:** Pippa K. Bailey, Adarsh Babu.

**Resources:** Pippa K. Bailey, Liise K. Kayler.

**Software:** Pippa K. Bailey, Liise K. Kayler.

**Supervision:** Yoav Ben-Shlomo, Fergus J. Caskey, Lucy E. Selman.

**Writing – original draft:** Pippa K. Bailey.

**Writing – review & editing:** Yoav Ben-Shlomo, Fergus J. Caskey, Mohammed Al-Talib, Hannah Lyons, Adarsh Babu, Liise K. Kayler, Lucy E. Selman.

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
