## [Decision Letter · Decision Letter 0]

22 Apr 2021

PONE-D-21-09369

Development of an intervention to improve AccesS to living-donor Kidney transplantation (The ASK study): a UK qualitative interview study with mixed stakeholders using the Person-Based Approach

PLOS ONE

Dear Dr. Bailey,

Thank you for submitting your manuscript to PLOS ONE. After careful consideration, we feel that it has merit but does not fully meet PLOS ONE’s publication criteria as it currently stands. Therefore, we invite you to submit a revised version of the manuscript that addresses the points raised during the review process.

This is an important piece of work that has generated interest of the reviewers given the timely and relevant nature of the topic, i.e. interventions to address socio-economic inequity in access to LD KTx. The expert reviewers came up with substantive comments and suggestions to improve the paper, and I agree with their assessments. The MS would be better if more concise (including the title as suggested by reviewer 2) as it is indeed extensive, and please check all figures and tables (as there had been issues with them upon primary submission). There might be selection bias at various levels, highlighted by 2 reviewers. Please address all reviewers' comments in a point-by-point discussion and make revisions accordingly. Beware that this invitation for major revisions does not automatically imply acceptance of the revised paper, as it will undergo thorough peer review again. Looking forward to receiving your revised manuscript.

We look forward to receiving your revised manuscript.

Kind regards,

Frank JMF Dor, M.D., Ph.D., FEBS, FRCS

Academic Editor

PLOS ONE

Journal Requirements:

Reviewers' comments:

Reviewer's Responses to Questions

**Comments to the Author**

1. Is the manuscript technically sound, and do the data support the conclusions?

Reviewer #1: Partly

Reviewer #2: Yes

Reviewer #3: Yes

2. Has the statistical analysis been performed appropriately and rigorously? 

Reviewer #1: N/A

Reviewer #2: N/A

Reviewer #3: Yes

3. Have the authors made all data underlying the findings in their manuscript fully available?

Reviewer #1: No

Reviewer #2: Yes

Reviewer #3: Yes

4. Is the manuscript presented in an intelligible fashion and written in standard English?

Reviewer #1: Yes

Reviewer #2: Yes

Reviewer #3: Yes

5. Review Comments to the Author

Reviewer #1: Dr Bailey presented on behalf of the research group their manuscript entitled Development of an intervention to improve AccesS to living-donor Kidney transplantation (The ASK study): a UK qualitative interview study with mixed stakeholders using the Person-Based Approach.

The matter of this investigation is relevant and contemporary. This is a qualitative research. The authors give the impression they are deeply familiarized with the methodology applied. They compared three different approaches to improve living kidney donation rates, interviewing patients, relatives and health care professionals. They aimed to design an intervention to enhance the chances of socioeconomically deprived end-stage renal disease patients’ access to the kidney transplant list. They propose to apply this instrument adapted to the UK population in a future RCT.

My concerns are the following:

-the manuscript is very extensive, including over 25 years old references (for example ref 1-3;10). The reader may get overwhelmed with the extensive details that are not essential to the research. I suggest being more concise.

-selection of participants seems biased. On page 9 is stated that the eligible population was identified by the local site primary investigators. Healthcare practitioners were contacted by e-mail based of the PI knowledge. My concern is how can be assured that these 12 patients and 4 relatives represents the variety of cases attending the two UK hospitals involved in this research. It is quite possible that introvert subjects were not selected, and other ethnic groups were underrepresented. Actually, 5 participants belong to other ethnic group, but it was not stated are they patients or healthcare providers. This is a major obstacle to the aim of enhancing socioeconomically deprived end-stage renal disease patients’ access to the kidney transplantation, and from my opinion lowers the validity of the results.

-participants selection flow chart is not shown. Actually, figures were not provided.

-it is not clearly exposed why 50% of the participants were healthcare professionals and 38% patients. In my understanding, patients’ and relatives’ opinion should be more extensively represented, clearly more than healthcare professionals. Patients and relatives can propose a model, and healthcare professionals evaluate feasibility.

-were the 12 patients interviewed already on the transplant list, or under evaluation? Did they have previous knowledge about LD?

-the think aloud technique is interesting, because it encourages proposing anything that come to the participant´s mind about a task. Would it be possible that UK patients could come with a new proposal, different from the three foreign models presented to them?

Reviewer #2: I have reviewed the manuscript entitled Development of an intervention to improve Access to living-donor Kidney transplantation (the ASK study): a UK qualitative interview study with mixed stakeholders using the Person-Based Approach. Due to evidence of socioeconomic inequity to live donor kidney transplant this study aims to develop a UK-specific intervention to support eligible persons to access a living kidney transplant. Three existing interventions from the literature were identified The Norwegian model, The home education model and the transplant candidate advocate model. The person-based approach was used for intervention development. Qualitative in-depth interviews were performed with persons with advanced kidney disease, their family members and transplant healthcare professionals to understand how different people view the proposed intervention components. The interviews were analysed with an inductive thematic analysis. Four general themes were identified and for each of the three intervention discussed three themes were identified. The results are illustrated with demographics of participants, a figure of the themes generated from the interviews as well as quotations in the body text to strengthen the themes. With the results a multi component intervention was developed suitable for the UK context. This intervention will be evaluated in a coming study.

This is a very well written manuscript and an important subject as well as interesting. A lot of effort has been put into making this thorough and detailed qualitative study. However, as reviewer I have a few minor comments and suggestions to improve the manuscript.

Title: It is a very long title, wouldn’t this be enough? Development of an intervention to improve Access to living-donor Kidney transplantation (the ASK study)

Study population: Page 9 line 173 You wrote: People with advanced kidney disease CKD 4-5 and in this you include patients with a functioning kidney transplant? I do not agree that this is correct.

Page 10 line 215: A query £20 voucher was given to the participants. Did the participants know in advanced that they would get the voucher after being interviewed? If Yes is that ethically correct? If no, please add the information in the text.

Results: You have a qualitative approach however in the results you describe parts numerically e.g. page 15 line 305 and 317 (7/32) and 11/32) respectively. This continues through to page 20. Usually qualitative results are not described numerically, please change.

I am also doubtful about the information given about each study participant that you quote, i.e. patient/gender/ age/education/occupational status. Is all this information needed for each quotation? I suggest you use only the first two or possibly the first three or explain why all are needed.

On page 24 line 526 the sentence; Only 9/32 participants expressed positive views about the use of advocates. Again, this is a numeric way of describing the result and also valuing by saying only 9/32: Firstly it is a qualitative study and secondly I am not sure about if it is only either, 9/32 is 28 percent - close to one third of the study population. However this is not a quantitative study, please reword this sentence to a qualitative style.

Reviewer #3: Reviewer Comments:

Page 3, line 69: the authors refer to “Figure 1-flow chart illustrating programme of research” but figure 1 does not seem to be part of the manuscript.

Page 3, line 75: reference is not provided for the definition for social support while references are provided for the other terms.

Page 4, line 80: authors refer to “Figure 2-Mediators of socioeconomic inequity in living-donor kidney transplantation” but figure 2 does not seem to be part of the manuscript.

Page 4, lines 86-88: The following sentence about “if low levels of patient activation contribute to difficulties ...” should be reworded as it is not currently easy to comprehend.

Page 6, line 139-140: The authors refer to using “a theory and evidence based approach (basinginterventions on published research evidence and existing theories”. Is the theory-based approach referring to the original theoretical frameworks/constructs that were used during the development of the three previously existing interventions (e.g., Norway model...) or is this referring to the development of the multicomponent intervention? Please clarify and elaborate on the specifics of the theoretical frameworks utilized.

Page 8-9 (Study Population): Did the study interview family members that were related to people with advanced kidney disease that also participated in the interviews?

Page 9, line 183-184: What does “socioeconomic status (sampled by healthcare professionals’ knowledge of patient’s education...” mean? Was socioeconomic status used for purposive sampling determined by healthcare professional’s knowledge of a patient’s educational attainment?

Page 13, line 265: The authors refer to “Figure 3-thematic diagram”, figure 3 seems to be missing.

Page 14, line 280: The number of participants that brought up the four general themes are presented for three sections not including perceived cultural norms. How many participants brought up perceived cultural norms?

Page 15, line 312: the employment status is missing from the participant.

It is recommended that the paper be proofread for typos.

6. PLOS authors have the option to publish the peer review history of their article (what does this mean?). If published, this will include your full peer review and any attached files.

Reviewer #1: No

Reviewer #2: No

Reviewer #3: No

---

## [Author Response · Author response to Decision Letter 0]

24 May 2021

Response to reviewers’ comments:

Formatting and data availability statement:

Authors’ response: We have reformatted the manuscript so that it meets PLOS ONE’s style requirements. We have updated the Data Availability statement as follows:

'This study generated qualitative data in the form of digital audio recordings and transcripts from interviews. Participants were asked to provide written consent to share their anonymised data with other researchers. Data will be shared only if consent has been provided. 29 of 32 participants provided consent for data sharing. Anonymised interview transcripts have been uploaded to the University of Bristol’s Research Data Repository: https://data.bris.ac.uk/data/. Audiofiles of the recorded interviews are not suitable for sharing as they carry a high risk of allowing the research participant to be identified, and the content of interviews includes sensitive information. 

Individuals who wish to access the dataset can contact the researchers directly or actively search the University of Bristol’s data repository. Although the qualitative transcripts have been anonymised, as personal and sensitive issues have been discussed we cannot rule out the risk of identification, and therefore access to these transcripts is controlled. Individual researchers will need to request access to the controlled data through the University of Bristol via the Data Access Committee (DAC) for approval, before data can be shared after their host institution has signed a Data Access Agreement. The procedure for accessing data can be found here: https://www.bristol.ac.uk/staff/researchers/data/accessing-research-data/.'

Reviewer #1: 

Dr Bailey presented on behalf of the research group their manuscript entitled Development of an intervention to improve AccesS to living-donor Kidney transplantation (The ASK study): a UK qualitative interview study with mixed stakeholders using the Person-Based Approach.

The matter of this investigation is relevant and contemporary. This is a qualitative research. The authors give the impression they are deeply familiarized with the methodology applied. They compared three different approaches to improve living kidney donation rates, interviewing patients, relatives and health care professionals. They aimed to design an intervention to enhance the chances of socioeconomically deprived end-stage renal disease patients’ access to the kidney transplant list. They propose to apply this instrument adapted to the UK population in a future RCT.

Authors’ response: Thank you for taking the time to read our manuscript and for your comments. The study team indeed comprises individuals with expertise in qualitative research, intervention development, complex intervention and pragmatic RCTs.

My concerns are the following:

-the manuscript is very extensive, including over 25 years old references (for example ref 1-3;10). The reader may get overwhelmed with the extensive details that are not essential to the research. I suggest being more concise.

Authors’ response: Thank you. We have substantially edited the manuscript and removed text not essential to the research. We have also removed older references where possible.

-selection of participants seems biased. On page 9 is stated that the eligible population was identified by the local site primary investigators. Healthcare practitioners were contacted by e-mail based of the PI knowledge. My concern is how can be assured that these 12 patients and 4 relatives represents the variety of cases attending the two UK hospitals involved in this research. It is quite possible that introvert subjects were not selected, and other ethnic groups were underrepresented. Actually, 5 participants belong to other ethnic group, but it was not stated are they patients or healthcare providers. This is a major obstacle to the aim of enhancing socioeconomically deprived end-stage renal disease patients’ access to the kidney transplantation, and from my opinion lowers the validity of the results.

Authors’ response: Selection bias is a quantitative research term, not usually applicable to qualitative research theory or methodology. 

Sampling needs to be consistent with a study’s aims. In most quantitative studies sampling aims to achieve a population representative sample, such that findings may be generalisable to the population from which participants were sampled. In this qualitative research study, sampling was purposive, with the aim of achieving diversity in personal characteristics, experiences and perspectives. Diversity was achieved as indicated by:

• Results Table 4 (page 13 clean document)– participants were recruited from different stakeholder groups, hospitals (transplant centre and non-transplanting referral centre), sexes, ages groups, ethnic groups, and differed with respect to their marital status, levels of education and employment status. By providing full details of the sample we enable readers to judge transferability to other settings and contexts.

• Diverse perspectives were disclosed in interviews as indicated in the results – a simple example is that some participants expressed positive views towards different intervention components whilst others expressed concerns or negative views, but the results detail differing views from participants.

• Theme saturation – an iterative approach was taken to sampling, data collection and analysis until theoretical data saturation was reached, and then sampling was stopped. 

The eligible population was identified by local site primary investigators as is required by NHS Research ethics (no identifiable information can be shared with researchers until a patient has provided consent). The local investigators were told who was eligible (age 18 years or older, English speaking, CKD 4 or 5, dialysis, transplant). Local site investigators (one of whom was the Chief Investigator PKB) then invited participants who were diverse in the characteristics specified in the manuscript. As interviews progressed, if we wanted to interview more women or younger people, for example, site investigators were asked to invite eligible individuals from these groups.

This study aimed to develop an intervention to improve access to living-donor kidney transplantation that targets barriers experienced by socioeconomically deprived individuals. We have undertaken research to investigate ethnic inequity [References i) Wong K et al. Investigating Ethnic Disparity in Living-Donor Kidney Transplantation in the UK: Patient-Identified Reasons for Non-Donation among Family Members. J Clin Med 2020;9(11):3751, ii) Bailey PK et al. Beliefs of UK Transplant Recipients about Living Kidney Donation and Transplantation: Findings from a Multicentre Questionnaire-Based Case-Control Study. J Clin Med 2019;9(1):31]. Whilst there is evidence of some confounding with ethnicity and socioeconomic deprivation the barriers that explain ethnic inequity in living-donor kidney transplantation are different to those experienced by those who are socioeconomically deprived across all ethnic groups.

If this intervention proves feasible to deliver and acceptable we aim to adapt it to target barriers identified in our previous work with respect to ethnic inequity. This will require formal cultural adaptation, via interviews with individuals from target ethnic minority groups (in the UK primarily Black/African/Caribbean/Black British). The following text is included in our discussion limitations section:

‘This intervention has been developed to particularly address variables that mediate socioeconomic inequity and may not address barriers that explain ethnic inequity in access to living-donor transplantation (reference in manuscript), which differ from those being targeted here(reference in manuscript). If the intervention proves acceptable, feasible and effective it will require formal adaptation for ethnic minority and non-English speaking groups. This will require adaptation beyond simple translation of language, and will require cultural adaptation, as well as specifically addressing different barriers to living-donor transplantation. Such cultural adaptation is required when evaluating an intervention in any new population: as illustrated in this study, interventions used in other countries do not automatically translate to a UK population.’ (page 32, lines 677-686, clean document).

-participants selection flow chart is not shown. Actually, figures were not provided.

Authors’ response: There is no participant selection flow chart. Participant selection flow charts are not a feature of qualitative research as the aim is not to achieve a population representative sample. Rather, as explained above with purposive sampling we aimed to ensure diversity in participant characteristics and demographics and diversity in views and perspectives.

-it is not clearly exposed why 50% of the participants were healthcare professionals and 38% patients. In my understanding, patients’ and relatives’ opinion should be more extensively represented, clearly more than healthcare professionals. Patients and relatives can propose a model, and healthcare professionals evaluate feasibility.

Authors’ response: Following submission of our manuscript we noted a mistake in Table 4 which we have now corrected. 13 participants were patients, 4 family members and 15 healthcare professionals. Therefore 47% of participants were healthcare professionals and 53% were patients or relatives.

Healthcare professionals are important stakeholders to interview as they will be required to a) deliver an intervention, b) recruit to and deliver research to evaluate the intervention, c) accommodate an intervention in existing care pathways, and d) manage the impact of a successful intervention in terms of increased LDKT assessment workload. As individuals with the capacity to facilitate or obstruct intervention delivery and evaluation, the views of healthcare practitioners are crucial to optimising an intervention before trial.

The main aim of this study was not to generate new ideas for interventions. As illustrated in Figure 1, this study followed our previous mixed-methods work to identify barriers to living-donor transplantation and to identify interventions that address these barriers. We aimed to investigate stakeholder views on models identified in the literature that target the barriers identified in our previous mixed-methods work (a ‘theory and evidence based approach’ to intervention development [Reference: O'Cathain A, Croot L, Duncan E, Rousseau N, Sworn K, Turner K, et al. Guidance on how to develop complex interventions to improve health and healthcare. BMJ Open. 2019;9:e029954.). After discussing the proposed intervention components all participants were asked if they had any other suggestions or ideas (please see ‘Other suggestions: Is there another approach you think might be helpful?’ in the S1 File. Example topic guide).

-were the 12 patients interviewed already on the transplant list, or under evaluation? Did they have previous knowledge about LD?

Authors’ response: The patient participants included people with chronic kidney disease stages 4 and 5 and those receiving kidney replacement therapy, including individuals receiving dialysis and individuals with a kidney transplant. All individuals who did not have a functioning transplant were eligible for transplantation and/or undergoing assessment for suitability. 

Assessment of knowledge about LD was not an aim of this study but all patient participants were aware of living donor transplants as a theoretical treatment option. To clarify this we have added a sentence to the methods: ‘All participants were aware of living-donor kidney transplantation as a theoretical treatment option for advanced kidney disease.’ (page 9, lines 181-182 clean)

-the think aloud technique is interesting, because it encourages proposing anything that come to the participant´s mind about a task. Would it be possible that UK patients could come with a new proposal, different from the three foreign models presented to them?

Authors’ response: The think-aloud technique is a method used to gather data when evaluating products, resources or tasks. It is not a technique employed without a focus for the ‘thinking aloud’. As indicated in the methods section, in this study think-aloud interviews were used when participants were reviewing drafted intervention resources (letters to family/friends, information leaflets, animations). Therefore participants could suggest changes to the resources (which could have included not using them at all) and they could suggest alternatives to the resources they were reviewing but not new models. As indicated above participants could suggest alternative models/interventions, outside the think-aloud questions (please see ‘Other suggestions: Is there another approach you think might be helpful?’) in the S1 File. Example topic guide.

Reviewer #2: 

I have reviewed the manuscript entitled Development of an intervention to improve Access to living-donor Kidney transplantation (the ASK study): a UK qualitative interview study with mixed stakeholders using the Person-Based Approach. Due to evidence of socioeconomic inequity to live donor kidney transplant this study aims to develop a UK-specific intervention to support eligible persons to access a living kidney transplant. Three existing interventions from the literature were identified The Norwegian model, The home education model and the transplant candidate advocate model. The person-based approach was used for intervention development. Qualitative in-depth interviews were performed with persons with advanced kidney disease, their family members and transplant healthcare professionals to understand how different people view the proposed intervention components. The interviews were analysed with an inductive thematic analysis. Four general themes were identified and for each of the three intervention discussed three themes were identified. The results are illustrated with demographics of participants, a figure of the themes generated from the interviews as well as quotations in the body text to strengthen the themes. With the results a multi component intervention was developed suitable for the UK context. This intervention will be evaluated in a coming study.

This is a very well written manuscript and an important subject as well as interesting. A lot of effort has been put into making this thorough and detailed qualitative study. However, as reviewer I have a few minor comments and suggestions to improve the manuscript.

Authors’ response: Thank you for taking the time to read our manuscript and for your suggestions for how to improve the manuscript.

Title: It is a very long title, wouldn’t this be enough? Development of an intervention to improve Access to living-donor Kidney transplantation (the ASK study)

Authors’ response: Yes, you’re quite right! We’ve edited the title as you have suggested. Thank you.

Study population: Page 9 line 173 You wrote: People with advanced kidney disease CKD 4-5 and in this you include patients with a functioning kidney transplant? I do not agree that this is correct.

Authors’ response: We don’t include people with a functioning kidney transplant in CKD 4-5. Individuals with a functioning transplant were included as individuals receiving kidney replacement therapy. We’ve now clarified the text to read:

‘People with advanced kidney disease (including i) individuals with Chronic Kidney Disease stages 4 and 5, and ii) individuals those receiving kidney replacement therapy (dialysis or a functioning kidney transplant))’. (page 9, lines 164-166., clean document).

Page 10 line 215: A query £20 voucher was given to the participants. Did the participants know in advanced that they would get the voucher after being interviewed? If Yes is that ethically correct? If no, please add the information in the text.

Authors’ response: Participants did know in advance that they would get a voucher after being interviewed. The participant information sheet stated that ‘We will refund any travel expenses. You will be given a £20 voucher to thank you for your time.’ This practice is recommended in national research guidance (e.g. from NIHR https://www.nihr.ac.uk/documents/payment-guidance-for-researchers-and-professionals/27392#Good_practice_for_payment_and_recognition_%E2%80%93_things_to_consider) to ensure patient contributors to research are not financially worse off as a result of research participation, and as a form of recognition and thanks for their time and contribution. This in part can help to achieve socioeconomic equity in research participation. This was approved by the NHS Research Ethics Committee and Health Research Authority prior to the study starting. 

Results: You have a qualitative approach however in the results you describe parts numerically e.g. page 15 line 305 and 317 (7/32) and 11/32) respectively. This continues through to page 20. Usually qualitative results are not described numerically, please change.

Authors’ response: As requested we have removed the numbers.

I am also doubtful about the information given about each study participant that you quote, i.e. patient/gender/ age/education/occupational status. Is all this information needed for each quotation? I suggest you use only the first two or possibly the first three or explain why all are needed.

Authors’ response: We have deleted the education level to reduce the information provided. We feel that the remaining information is required to provide a) evidence that the quotes are from different participants, and diverse participants, particularly with respect to socioeconomic status given the aim of this study, and b) context for some of the comments (e.g. one participant talks about not being able to take time off work to engage with a home education visit). 

On page 24 line 526 the sentence; Only 9/32 participants expressed positive views about the use of advocates. Again, this is a numeric way of describing the result and also valuing by saying only 9/32: Firstly it is a qualitative study and secondly I am not sure about if it is only either, 9/32 is 28 percent - close to one third of the study population. However this is not a quantitative study, please reword this sentence to a qualitative style.

Authors’ response: Thank you. As the preceding sentence contains the same information we have deleted the sentence you have highlighted. This section now reads: ‘The transplant candidate advocate intervention component was the least popular option, with most participants expressing negative views towards it or concerns about its use.’

Reviewer #3: 

Page 3, line 69: the authors refer to “Figure 1-flow chart illustrating programme of research” but figure 1 does not seem to be part of the manuscript.

Authors’ response: Please accept our apologies. Figures were uploaded late, after initial submission. Figure 1 has now been provided in the revised manuscript.

Page 3, line 75: reference is not provided for the definition for social support while references are provided for the other terms.

Authors’ response: A reference was provided for the definition of social support: Langford C, Bowsher J, Maloney J, Lillis P. Social support: a conceptual analysis. Journal of Advanced Nursing. 1997;25:95-100.

Page 4, line 80: authors refer to “Figure 2-Mediators of socioeconomic inequity in living-donor kidney transplantation” but figure 2 does not seem to be part of the manuscript.

Authors’ response: Please accept our apologies. Figures were uploaded late, after initial submission. Figure 2 has now been provided in the revised manuscript.

Page 4, lines 86-88: The following sentence about “if low levels of patient activation contribute to difficulties ...” should be reworded as it is not currently easy to comprehend.

Authors’ response: We have reworded this sentence to make it easier to understand:

‘For example, if low levels of patient activation mean that a patient find it difficult to approach potential donors, a ‘work-around’ solution would be for a healthcare practitioner to approach potential donors on a patient’s behalf.’ (page 4, lines 85-88, clean document).

Page 6, line 139-140: The authors refer to using “a theory and evidence based approach (basing interventions on published research evidence and existing theories”. Is the theory-based approach referring to the original theoretical frameworks/constructs that were used during the development of the three previously existing interventions (e.g., Norway model...) or is this referring to the development of the multicomponent intervention? Please clarify and elaborate on the specifics of the theoretical frameworks utilized.

Authors’ response: The ‘theory and evidence based approach’ refers to the development of the multicomponent intervention. To clarify this we have now re-written the relevant section so that it now reads:

‘We undertook intervention development using an approach that was both ‘theory and evidence based’ and ‘target population centred’ (34). ‘Theory and evidence based’ approaches develop interventions by combining published research evidence and existing theories. As indicated above we identified three existing intervention components from the published literature. One intervention (Home-based patient and family education) had been developed with respect to multisystemic therapy theory and had RCT evidence of effectiveness. One intervention (TCAs) had weak evidence of effectiveness from an observational study, and one (‘the Norway model’) had not been formally evaluated in research. ‘Target population centred’ approaches develop interventions based on the views of the people who will use them, and we employed the ‘Person-Based Approach’ (PBA) to do this.’ (page 7, lines 136-145, clean document).

Page 8-9 (Study Population): Did the study interview family members that were related to people with advanced kidney disease that also participated in the interviews?

Authors’ response: One of the family participants was related to one of the participants with advanced kidney disease. We have added a sentence explaining this to the results.

Page 9, line 183-184: What does “socioeconomic status (sampled by healthcare professionals’ knowledge of patient’s education...” mean? Was socioeconomic status used for purposive sampling determined by healthcare professional’s knowledge of a patient’s educational attainment?

Authors’ response: Yes, socioeconomic status was used for purposive sampling. Sampling was based on several measures of socioeconomic status including patient’s education level, employment status and housing/postcode. We agree that the used of the words ‘by healthcare professionals’ knowledge’ is confusing and so we have edited this sentence to now read:

‘Purposive sampling of patient participants was undertaken to ensure diversity in sex, age, ethnicity, and socioeconomic status (using the following socioeconomic measures: patient’s education level, employment status and housing/postcode).’ (page 9, lines 174-176, clean document).

Page 13, line 265: The authors refer to “Figure 3-thematic diagram”, figure 3 seems to be missing.

Authors’ response: Please accept our apologies. Figures were uploaded late, after initial submission. Figure 3 has now been provided in the revised manuscript.

Page 14, line 280: The number of participants that brought up the four general themes are presented for three sections not including perceived cultural norms. How many participants brought up perceived cultural norms?

Authors’ response: 16/32 participants mentioned societal or cultural norms. At the request of Reviewer 2 we have now removed the numbers presented as distracting quantitative information.

Page 15, line 312: the employment status is missing from the participant.

Authors’ response: This participant is a healthcare practitioner ‘Home dialysis nurse’. Employment status is only provided for patient and family participants.

It is recommended that the paper be proofread for typos.

Authors’ response: Thank you. We have done this.

---

## [Decision Letter · Decision Letter 1]

10 Jun 2021

Development of an intervention to improve AccesS to living-donor Kidney transplantation (The ASK study)

PONE-D-21-09369R1

Dear Dr. Bailey,

We’re pleased to inform you that your manuscript has been judged scientifically suitable for publication and will be formally accepted for publication once it meets all outstanding technical requirements.

Kind regards,

Frank JMF Dor, M.D., Ph.D., FEBS, FRCS

Academic Editor

PLOS ONE

Additional Editor Comments (optional):

Reviewers' comments:

Reviewer's Responses to Questions

**Comments to the Author**

1. If the authors have adequately addressed your comments raised in a previous round of review and you feel that this manuscript is now acceptable for publication, you may indicate that here to bypass the “Comments to the Author” section, enter your conflict of interest statement in the “Confidential to Editor” section, and submit your "Accept" recommendation.

Reviewer #1: All comments have been addressed

Reviewer #2: All comments have been addressed

Reviewer #3: All comments have been addressed

2. Is the manuscript technically sound, and do the data support the conclusions?

Reviewer #1: Yes

Reviewer #2: (No Response)

Reviewer #3: Yes

3. Has the statistical analysis been performed appropriately and rigorously? 

Reviewer #1: N/A

Reviewer #2: (No Response)

Reviewer #3: Yes

4. Have the authors made all data underlying the findings in their manuscript fully available?

Reviewer #1: Yes

Reviewer #2: (No Response)

Reviewer #3: Yes

5. Is the manuscript presented in an intelligible fashion and written in standard English?

Reviewer #1: Yes

Reviewer #2: (No Response)

Reviewer #3: Yes

6. Review Comments to the Author

Reviewer #1: Dr Bailey has elegantly answered all the queries. There is a better description of the methodology used to recruit participants. As in qualitative research is usually done, in the clean version the results that were described numerically were removed and the results are expressed in a suitable way. The authors still did not provide data on the patients with CKD being listed for KT or even being considered as suitable candidates for KT. However the authors now included a sentence stating that the patients were aware of KT and LD in particular.

I consider that the manuscript has been remarkably improved and achieved good quality to be considered for publication.

Reviewer #2: (No Response)

Reviewer #3: (No Response)

7. PLOS authors have the option to publish the peer review history of their article (what does this mean?). If published, this will include your full peer review and any attached files.

Reviewer #1: No

Reviewer #2: No

Reviewer #3: No

---

## [Editor Report · Acceptance letter]

15 Jun 2021

PONE-D-21-09369R1 

Development of an intervention to improve AccesS to living-donor Kidney transplantation (The ASK study) 

Dear Dr. Bailey:

I'm pleased to inform you that your manuscript has been deemed suitable for publication in PLOS ONE. Congratulations! Your manuscript is now with our production department. 

Kind regards, 

on behalf of

Dr. Frank JMF Dor 

Academic Editor

PLOS ONE